# Do genetic ancestry tests increase racial essentialism? Findings from a randomized controlled trial

**Wendy D. Roth**[1,2]*, **Şule Yaylacı**[1,2,3], **Kaitlyn Jaffe**[2], **Lindsey Richardson**[2,4]

**1** Department of Sociology, University of Pennsylvania, Philadelphia, PA, United States of America, **2** Department of Sociology, University of British Columbia, Vancouver, BC, Canada, **3** Institute for European Studies, University of British Columbia, Vancouver, BC, Canada, **4** British Columbia Centre on Substance Use, Vancouver, BC, Canada

* wroth3@sas.upenn.edu

**Data Availability Statement:** The data used to create this analysis are available at this link in the Harvard Dataverse: https://doi.org/10.7910/DVN/DZ099T

## Abstract

Genetic ancestry testing is a billion-dollar industry, with more than 26 million tests sold by 2018, which raises concerns over how it might influence test-takers' understandings of race. While social scientists argue that genetic ancestry tests may promote an essentialist view of race as fixed and determining innate abilities, others suggest it could reduce essentialist views by reinforcing a view of race as socially constructed. Essentialist views are a concern because of their association with racism, particularly in its most extreme forms. Here we report the first randomized controlled trial of genetic ancestry testing conducted to examine potential causal relationships between taking the tests and essentialist views of race. Native-born White Americans were randomly assigned to receive Admixture and mtDNA tests or no tests. While we find no significant average effect of genetic ancestry testing on essentialism, secondary analyses reveal that the impact of these tests on racial essentialism varies by type of genetic knowledge. Within the treatment arm, essentialist beliefs significantly declined after testing among individuals with high genetic knowledge, but increased among those with the least genetic knowledge. Additional secondary analysis show that essentialist beliefs do not change based on the specific ancestries reported in test-takers' results. These results indicate that individuals' interpretations of genetic ancestry testing results, and the links between genes and race, may depend on their understanding of genetics.

## Introduction

Since the completion of the Human Genome Project in 2003, Direct-to-Consumer (DTC) genetic ancestry tests (GATs) have become a billion-dollar industry [1] with at least 74 companies selling tests online [2]. For a typical test, an individual submits a genetic sample from a cheek swab or saliva by mail and receives charts linking a particular family line to specific ancestry groups or regions and/or estimating what proportion of their ancestry is purportedly European, sub-Saharan African, Native American, or Asian–categories that map onto popular

**Funding:** This research was funded by grants from the Social Sciences and Humanities Research Council of Canada (#435- 2014-0467; WR), the Canada Foundation for Innovation (#23744; WR), and a UBC Killam Faculty Research Fellowship (WR). Kaitlyn Jaffe is supported by a Vanier Canada Graduate Scholarship from the Canadian Institutes of Health Research and by the University of British Columbia Public Scholars Initiative. Lindsey Richardson is supported by a Michael Smith Foundation for Health Research Scholar award and Canadian Institutes of Health Research New Investigator (MSH 217672) and Foundation (FDN-154320) awards." The funders and Family Tree DNA had no role in study design, data collection and analysis, decision to publish, or preparation of the manuscript.

**Competing interests:** The authors have declared that no competing interests exist.

conceptions of race. The industry has grown exponentially; more than 26 million tests had been sold by 2018, and more people took a test in 2018 than in all previous years combined [3]. There are many critiques of the legitimacy and reliability of GATs [4–6], yet their continued popularity calls for investigation of how these tests affect users' beliefs about race and their intergroup behaviors.

How these tests influence test-takers' understanding of race is of particular concern. While there are small genetic differences between populations that allow geneticists to trace their global migrations, these variations do not support the idea of discrete races that can be distinguished by genes alone [7–11]. Social scientists have long asserted that race is socially constructed, for instance with classifications and their meaning changing over time and place, even if race refers in part to biological or descent-based characteristics [12,13]. Yet a genetic essentialist view of race, that holds that genes alone determine races and imbue them with distinct and immutable essences that are associated with different skills or abilities [12,13], continues to exist among the general public [13,14]. Genetic essentialism is one type of essentialism which locates the group's core essence in its genes rather than elsewhere (e.g., in the group's culture, as in cultural essentialism) [15]. One implication of this view is that discrete biological races exist within the human species and that genetic difference is the root of racial differences in behavior and outcomes [14]. Essentialist views of race have significant negative consequences for intergroup behavior, including less willingness to interact with other races, greater endorsement of racial stereotypes [16], and association with traditional and modern racism [14]. These beliefs have historically led to eugenicist movements, ethnic cleansing, apartheid, and genocide [12,14,17,18].

Some social scientists argue that GATs are likely to reinforce a genetic essentialist view of race [4,17,19,20]. Many people view scientific information as objective and authoritative and may therefore give it more consideration than social or environmental factors in understanding complex concepts [21–24]. The presentation of GAT results in the form of categories that overlap with commonly used racial groups reinforces a view that these tests report about race rather than biogeographical ancestry [24]. The methodology of ancestry testing itself may therefore reify races as essential genetic realities by suggesting that genetic tests can distinguish them [17].

Others have speculated that taking the tests could have the opposite effect by reinforcing belief in the socially constructed nature of race [25–28]. For example, a test-taker who receives evidence of an ancestry that is not part of her social experience of race may question the meaning of racial categories and recognize their foundation in social experience more than biology. Some journalistic accounts depict testing as reinforcing the views that all humans have mixed racial origins and that long-held identities are more than what genes reveal [29–31]. From this perspective, some question whether GATs might have the potential to break down racial categories and social distance [25–27].

The few studies seeking to adjudicate between these outcomes have focused on the effect of reading media articles about GATs and found that reading articles depicting the tests as able to reveal a person's race, or emphasizing the degree of overall genetic difference between groups, increases belief in essential racial and ethnic differences [17,32,33]. However, the media articles in these studies included clear statements supporting or opposing a genetic basis to race; by contrast, the experience of taking a GAT requires the test-taker to actively interpret complex, personalized results.

In this paper we ask: does taking GATs increase or decrease an essentialist view of race? The overall aim of this study is to adjudicate between these competing hypotheses, which requires a causal analysis of the effect of taking GATs on essentialist views with a randomized controlled trial (RCT). We present the findings of the first such RCT here.

In addition, we conduct secondary analyses to identify factors with plausible potential to influence the impact of GATs on genetic essentialism that could be valuably incorporated into subsequent experimental research. We anticipate that the interpretation of the results is a key mechanism influencing the impact these tests have on test-takers' understanding of race. Those interpretations may be influenced by the test-takers' genetic knowledge–their factual or scientific knowledge of basic genetics. For example, those with little genetic literacy may struggle to fully interpret their test results [34,35] and may be more likely to view them as supporting "lay theories" of essential genetic differences between races [36]. We compare changes over time for people with different levels of genetic knowledge in the control and treatment arms. It is also possible that the specific ancestry results people receive–specifically the "confirmation" or "discovery" of new ancestries–influence their belief in racial essentialism. We therefore conduct additional non-experimental analyses within the treatment arm to explore these possibilities.

## Overview of study goals and hypotheses

We conducted this RCT with a sample of 802 native-born White Americans who were willing to take GATs. Half were randomly assigned to receive commercially-available admixture and mtDNA tests purchased from Family Tree DNA, one of the largest DTC ancestry testing companies [37]. The other half received no tests, as our goal was to gauge the effect of taking GATs relative to those who do not. We developed a new scale to measure genetic essentialist beliefs about race, which includes nine items focusing on how people think about the relationship between genes and race, with four-point disagree/agree Likert scale response options. We included the scale items in both the pre-test and post-test surveys [15].

Given aforementioned experimental studies on the impacts of reading media articles about the tests [17,32,33], in primary analysis we hypothesize that taking GATs would increase levels of genetic essentialism. In secondary analyses, we hypothesize that the direction of the tests' effect on genetic essentialism will depend on the genetic knowledge of the test-taker since basic genetic knowledge is crucial for interpreting the results [34,35,38]. Many White Americans have limited genetic literacy [39]; even those with high educational attainment have difficulty understanding certain aspects of genetic information [40]. Research on health-focused genetic testing shows that many people incorrectly interpret the test results [41,42], and that lower levels of genetic literacy are associated with lower comprehension of results [34,35]. For ancestry tests as well, many test consumers have difficulty translating the information they receive into usable knowledge [1], a fact acknowledged by DTC companies [43]. Admixture test results often include a breakdown of geographic ancestral origins as percentages (S1 Fig), while mtDNA and Y-DNA test results often present a map of historic migration routes (S2 Fig). These results can be interpreted in different ways. For instance, the map depicting humanity as branching off the same origins and spreading across the globe may be perceived as evidence of one human race or as emphasizing distinctions based on where the routes end, depending on genetic literacy. Test-takers without a basic understanding of genetics (e.g. not aware that 99.9% of the human genetic code is identical in every person) are more likely to refer to "lay theories"–genetic explanations for perceived race differences [36]–in their interpretations, and read the biogeographical categories as races, which may foster more essentialist views. We therefore expect test-takers lacking basic genetic knowledge to develop more essentialist views after testing. By contrast, we expect test-takers with high knowledge of genetics to be more likely to read the results as evidence that races and racial traits are not determined solely by genetics, and thereby develop less essentialist views after testing.

In additional secondary analysis of data from treatment arm participants, we hypothesize a null main effect of "confirming" or "discovering" a specific ancestry, but predict that any impact of the specific admixture test results on test-takers' belief in genetic essentialism also varies by their type of genetic knowledge. To those with little understanding of genetics, test results "confirming" the ancestries respondents already know of may reinforce a belief that race is revealed in their genes [44]. "Discovering" a new ancestry unknown to respondents may also influence those test-takers' beliefs in essentialism. Receiving results reporting known or new ancestries may feel like confirmation or discovery to these test-takers, even though the results are easily misunderstood and their interpretations may be incorrect. We expect that those with weaker knowledge of genetics will be more likely to believe that the ancestries reported are their "true" race, increasing essentialism. By contrast, we expect that a stronger knowledge of genetics, and thus a greater ability to interpret the results, is likely to reduce any impact on essentialism that the test's "confirmation" of an existing identity or reporting of a new ancestry may otherwise have.

## Materials and methods

### Participants and procedure

Participants were recruited through random-digit dialing and screened for eligibility. Eligible individuals were those born in the U.S., aged 19 or older, who self-identified as non-Hispanic White, and where neither they nor any relatives had taken GATs but they were willing to take one. Study costs and sample sizes required for analysis necessitated focusing on a single population group; we focused on non-Hispanic Whites because our informal communications with several testing companies indicated that they were the largest U.S. consumer group of the tests [45]. Our target population is not all native-born non-Hispanic Whites, but those who were willing to take GATs and had not previously received personalized genetic ancestry information, a group for which we were unable to find representative data. While we cannot claim that our data represent the target population, we stratified our sampling by gender, age, education, and region based on the national population of native-born non-Hispanic Whites aged 19 and older to improve the demographic diversity of our sample beyond that of a convenience or regional sample. See S1 Appendix for a comparison of our data with existing data representing the larger population of non-Hispanic Whites.

Study protocols were approved by the University of British Columbia Behavioral Research Ethics Board (H14-02090). Written informed consent was obtained from each participant as part of the registration process. The final analytical sample consists of 802 participants (control N = 425; treatment N = 377; Fig 1). The sample is 36.7% male and 63.3% female. For age, 9.9% of participants are age 19–34, 37.3% are age 35–54, and 52.9% are 55 or older. For educational attainment, 10.7% of participants have a high school degree or less, 27.9% have some college education, 28.9% have a college degree, and 32.4% have more than a college education. S3 Table, column 1, shows the sample's demographic characteristics, while S2 Table, columns 5 and 6 ("Remaining Participants") shows descriptive statistics for all variables used in analysis separately for the control and treatment groups.

Participants completed an online pre-test survey between October 2014 and February 2015. After that, the treatment group received a test kit by mail with instructions to send a DNA sample to the testing company. We asked control participants not to take GATs and offered them a discount coupon to purchase the same tests at half price after the study. Admixture and mtDNA tests were conducted on the treatment group's samples and results were e-mailed to them. To ensure treatment participants engaged with the results, we asked them to spend at least 30 minutes reviewing the results online and then take a short "First

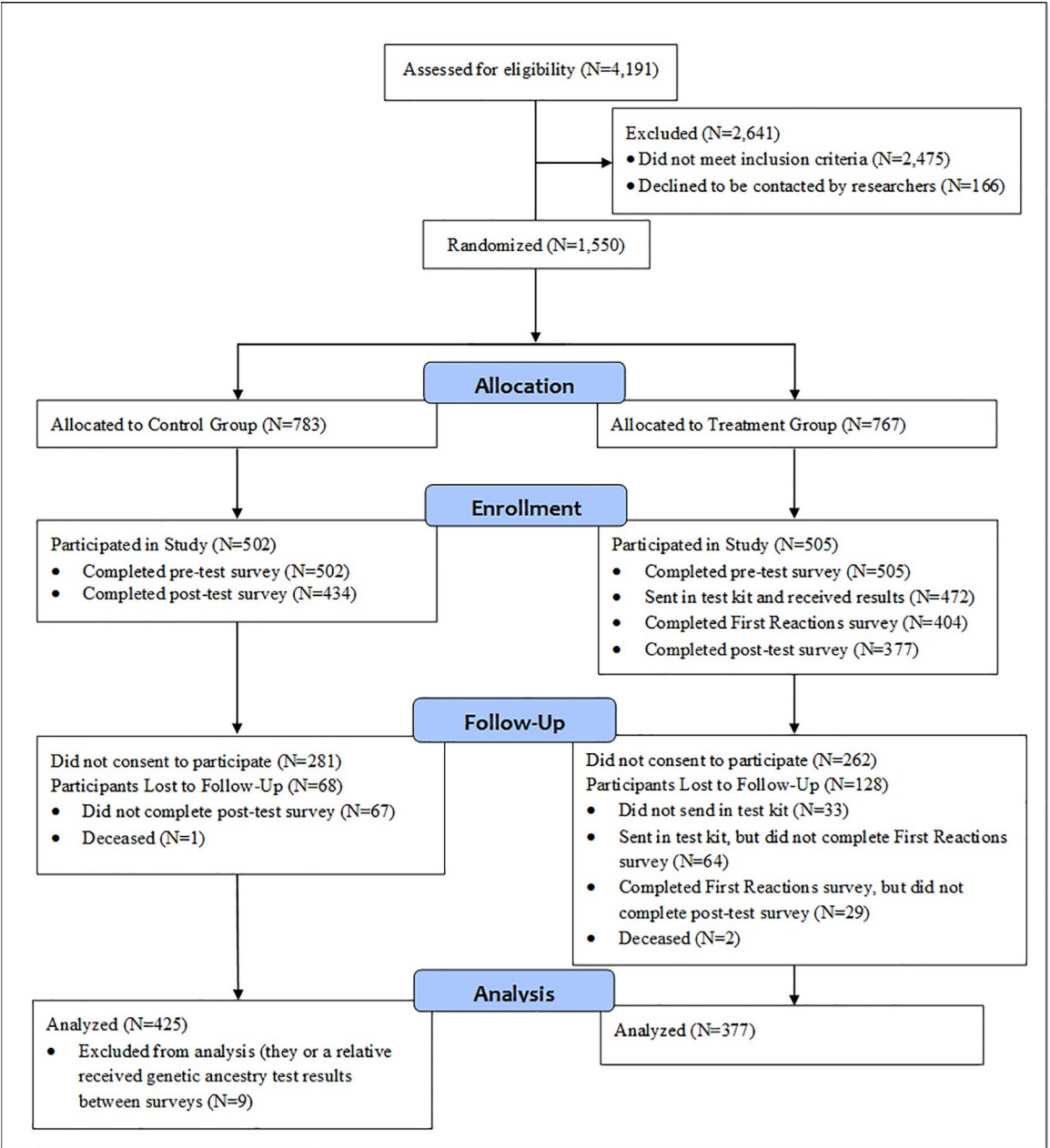

**Fig 1. CONSORT flow diagram of study design and response.**

Reactions" survey, conducted between Jan.-July 2015. A final online post-test survey was conducted between Sept. 2015-Mar. 2016 among all participants. We offered small financial incentives for completing each survey. Participants were invited to complete the post-test survey 11 months after completing the pre-test survey, and approximately eight months after treatment respondents received their results. This ensured that sufficient time had passed for the test-takers to cognitively absorb their results and potentially share the information with others; any observed changes are hence more likely to be lasting effects rather than immediate reactions.

## Measures

We operationalized genetic essentialism with a newly-developed scale, the Genetic Essentialism Scale for Race (GESR), using nine survey questions asked in both the pre- and post-test surveys [15]. The items focus on how people think about the relationship between genes and race, and have four-point disagree/agree Likert scale response options (S4 Table). Using exploratory and confirmatory factor analysis, we developed GESR as a second-order construct that incorporates several first-order factors comprising subsets of belief in genetic essentialism (i.e., the belief that races are discrete, immutable categories determined by genes; the belief that genes cause races to have distinct and innate essences associated with different traits, skills, or abilities; and the belief that, rather than sharing common roots, races evolved from two or more distinct ancestry groups) [15]. Our tests showed that the second-order factor, GESR, which incorporated these components into a broader concept of genetic essentialism had high validity and reliability (Composite Reliability (CR) = 0.838 and Average Variance Extracted (AVE) = 0.635; the thresholds for these measures are $> = 0.7$, and $> = 0.5$, respectively, and convergent validity is achieved when $CR > AVE$ [46–48]), whereas the individual first-order factors, representing only parts of the concept, did not perform as well. We also confirmed the construct validity of the GESR, hypothesizing and confirming that the higher the GESR, the higher the opposition to affirmative action policies. A detailed description of the development and validation of the GESR scale has been published previously [15]. The Cronbach's alpha coefficients for the 9 items in the scale for the current sample were 0.70 in the pre-test and 0.71 in the post-test. A lower score on the scale is consistent with a social constructionist view of race.

While there are different facets of genetic literacy, we focused on the factual knowledge dimension which is highest in the knowledge hierarchy of genomics [49] and most pertinent to the cognitive steps of moving from reading results to forming attitudes. To measure basic genetic knowledge, we used two pre-test survey questions drawn from the Survey on Genomics Knowledge, Attitudes and Policy Views (GKAP) [50]: 1) "Based on what you know, would you say that DNA can be found in every cell in the human body or only in specific organs and cells in the human body?" 2) "Based on what you know, would you say that more than half, about half, or less than half of a human being's genes are identical to those of a mouse?" (See S1 Appendix & S3 Table for comparison of our sample's responses to those of GKAP respondents). The GKAP and our survey also included a third item: "Based on what you know, would you say that more than half, about half, or less than half of a White person's genes are identical to those of a Black person?" We excluded this item from this analysis because of its association with racial essentialism. However, models that we ran which included this item showed similar results to those presented in the paper.

Because the questions vary in difficulty (91.86% answered the first correctly, 32.87% for the second), we weighted the responses based on the question's difficulty and how close respondents came to the correct answer. As a proxy for difficulty, we used percentages answering correctly and multiplied this by the scores assigned to the response options: one point for the correct answer (the first option in both), a half point for those closer to the correct answer on the second question ("about half"), and zero points for other responses, including "Don't Know" which we treated as equivalent to "No Knowledge" (S5 Table). After applying these weights, we constructed the genetic knowledge scale by adding individuals' response scores. This produced a raw scale with six points; however, two of the six points in the range had very few cases in them. We recoded these with the closest category, which yielded four effective categories. We use this variable as a 4-category ordinal measure of genetic knowledge, ranging from "No Knowledge" to "High Knowledge." While this measure best fits the pattern of

responses to the genetic knowledge questions, we were concerned that our findings might not be robust due to the sample sizes within these categories (in particular, the 'no knowledge' group has 31 and 21 cases in the control and treatment groups respectively). We therefore also created an alternative dichotomous variable that grouped 'no/low knowledge' and 'medium/ high knowledge' together (see S6 Table). We ran sensitivity analyses using this dichotomous version of the measure as well.

For the treatment group, we compared the ancestries reported in their admixture test results to the ethnic identities and ancestral origins listed in the pre-test survey. Because people may list only some ethnic identities from all the ancestries they know of [51], we asked them to list all their ethnic identities and also all the ancestral origins they knew of for each biological parent. We examined whether their admixture results "confirmed" any ancestry listed for themselves or their parents, and whether they reported any new ancestry that was not listed for themselves or their parents (see S1 Appendix for discussion of coding).

### Analyses

Analyses were conducted using Stata (Version 14.2; StataCorp) and SPSS Statistics (Version 25; IBM). We examined pre-treatment equivalence across study arms and assessed attrition patterns for systematic difference relevant to our hypotheses (presented in S1 Appendix). Following this, our first set of analyses uses linear mixed-effects models, to examine the causal effect of taking GATs on genetic essentialism, comparing control and treatment groups. In the text, we present our findings as average predicted probabilities of genetic essentialism estimated by the linear mixed-effects models (LMM). We supplement these analyses with Ordinary Least Squares (OLS) regression.

Next, using the same techniques, we conducted secondary analysis examining within-group changes in genetic essentialism for control and treatment groups, subdividing participants by baseline genetic knowledge to determine whether such knowledge was associated with changes in genetic essentialism. Our second set of analyses uses the 4-category ordinal measure of genetic knowledge, while our third set is a sensitivity analysis using the dichotomous measure of genetic knowledge.

Finally, our fourth set of analyses presents non-experimental results focusing only on the treatment group to examine the effect of specific GAT results on genetic essentialism, while controlling for and interacting test results with baseline genetic knowledge. We used OLS regression to examine the effects of the admixture test results 1) "confirming" an ancestry (reporting an ancestry of which the respondent had prior knowledge); 2) "discovering" a new ancestry (reporting an ancestry of which the respondent did not have prior knowledge); and 3) interacting with genetic knowledge. Ancestries are presented as European and non-European for clarity.

All models, both experimental and non-experimental, control for living in the South, interaction with non-Whites and political party preference, as well as gender, age, and education. Additional details are provided in S1 Appendix.

### Results

In our first set of analyses with LMM models comparing genetic essentialism between control and treatment groups, the results did not show any significant average difference between the pre-test and post-test essentialism scores of the control and treatment groups (Fig 2). The predicted probability of the control group's genetic essentialism score increases by 0.005 points while that of the treatment group decreases by 0.006, yet neither difference is statistically

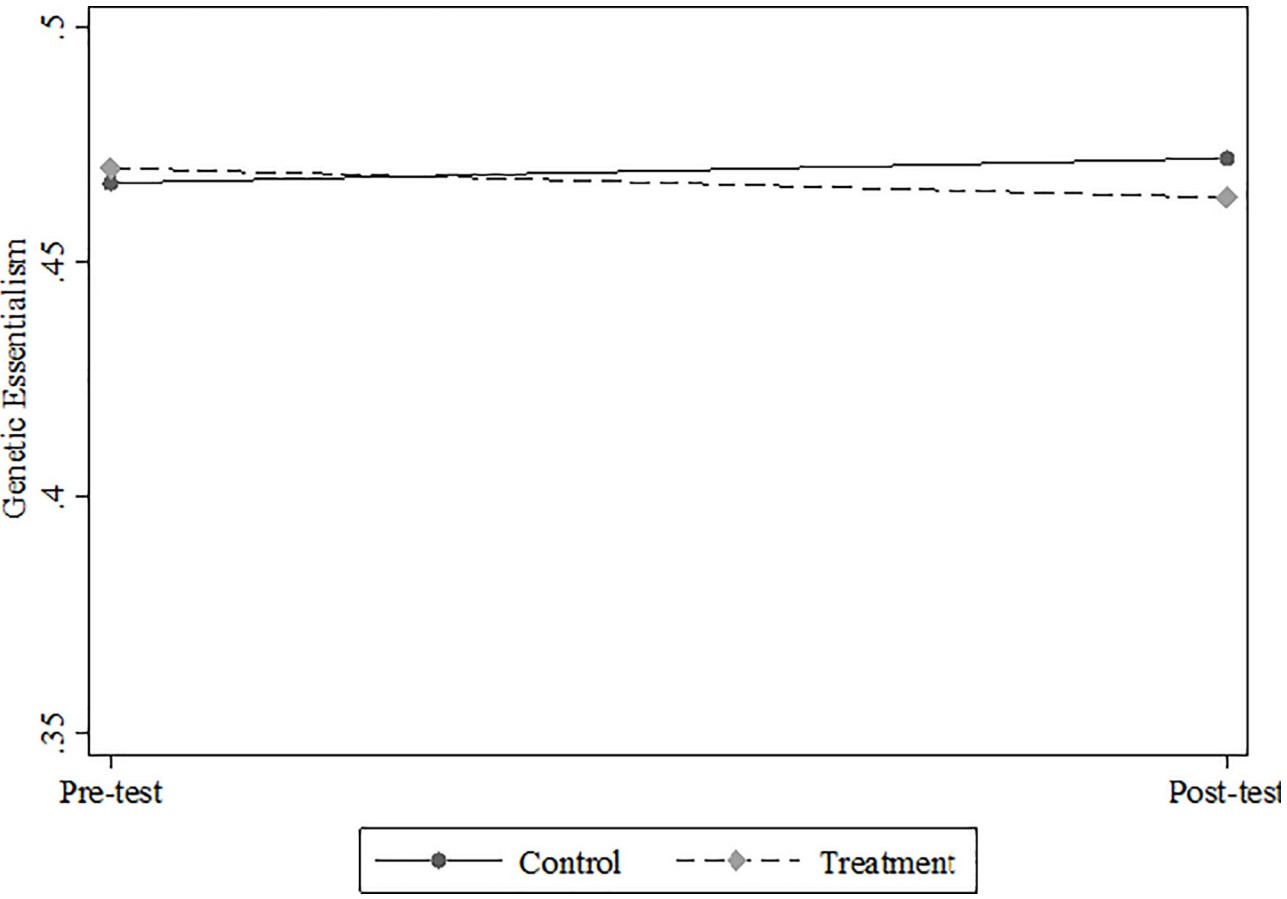

**Fig 2. Pre-test and post-test genetic essentialism for control and treatment groups (N = 794).** This graph plots the average marginal change in genetic essentialism scores of both the control and treatment groups. The pre-test genetic essentialism scores of the control and treatment groups are 0.468 and 0.469 respectively. The post-test genetic essentialism scores are 0.473 for the control group and 0.463 for the treatment group. We used Stata's "margins" command to produce the graph.

significant (p = 0.379 and p = 0.322, respectively). That is, contrary to our hypothesis, taking GATs did not impact genetic essentialist views on average.

However, our secondary analyses of within-group changes in genetic essentialism by baseline genetic knowledge found different patterns of change in genetic essentialism beliefs in the control and treatment arms, as depicted in Fig 3. Specifically, in our analyses using the 4-category ordinal measure of genetic knowledge, the no knowledge and high knowledge groups show contrasting changes between pre- and post-test in the treatment group but not in the control group. In the LMM, control respondents with high knowledge of genetics did not exhibit a significant change in the average predicted probabilities of their essentialism scores (Δ = -0.015; p = 0.157) and, similarly, control group respondents with no genetic knowledge did not show a significant change (Δ = 0.013; p = 0.543). Conversely, the average predicted probabilities for the treatment respondents with high knowledge of genetics declined significantly after taking the test (Δ = -0.040; p < 0.001). Treatment respondents with no knowledge of genetics on our scale showed an increase in their predicted essentialism scores after taking the test (Δ = 0.058; p = 0.026). That is, among participants who took genetic tests, belief in genetic essentialism decreased among those with high genetic knowledge and increased among those with no genetic knowledge (S9 Table). OLS regression models further confirmed

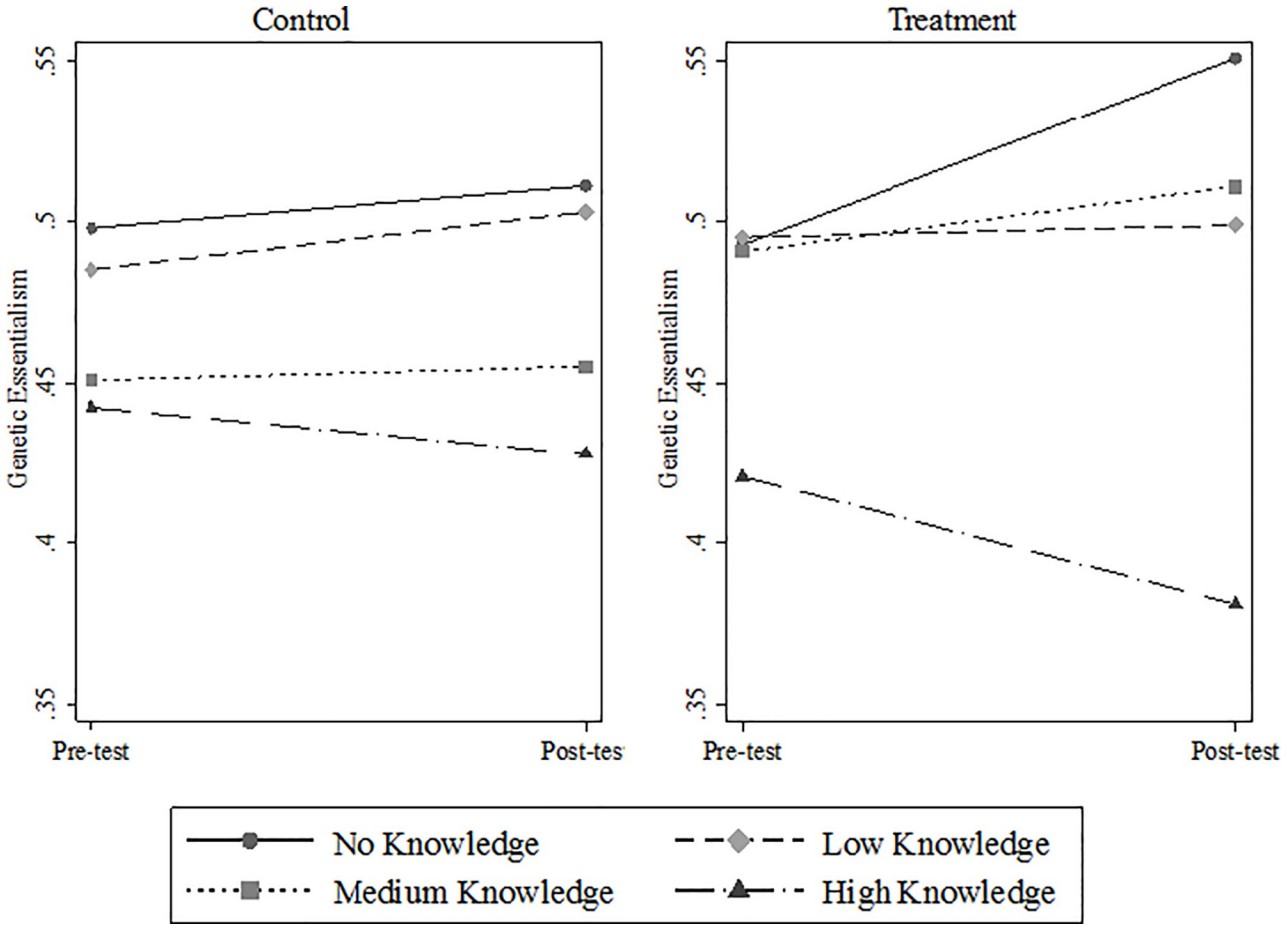

**Fig 3. Pre-test and post-test genetic essentialism by genetic knowledge (4-category ordinal) for control (N = 419) and treatment groups (N = 375).**

these interactions by showing that the difference between the treatment effect of taking GATs on respondents with no genetic knowledge ($\beta$ = 0.086) and with high genetic knowledge ($\beta$ = -0.032) was statistically significant (S10 Table).

In LMM models, the predicted average essentialism scores of control respondents with low genetic knowledge also increased ($\Delta$ = 0.018; p = 0.035), while there was no significant predicted change among treatment respondents with low knowledge ($\Delta$ = 0.004; p = 0.667). This increase in the control group may reflect changes in the U.S. political landscape between the survey waves. The low knowledge group may have been more susceptible to the rising salience of race-related discourse in the 2016 election campaigns. Yet, the treatment group, also subject to the same political changes, did not show similar increase, which we may read as the depressing effect of taking the tests.

LMM results from sensitivity analyses using the dichotomous measure genetic knowledge measure, show that average predicted probabilities of genetic essentialism declined significantly in the treatment group with higher genetic knowledge ($\Delta$ = -0.025; p = 0.007), while control respondents with higher knowledge did not show a significant change ($\Delta$ = -0.010; p = 0.280) (S11 Table and S3 Fig). There is a significant increase in the predicted average essentialism scores of control respondents with lower genetic knowledge ($\Delta$ = 0.017; p = 0.030),

while there is no significant predicted change among treatment respondents with lower knowledge (Δ = 0.009; p = 0.265).

In sum, in analyses examining changes in genetic essentialism with a specific focus on levels of genetic knowledge within the control and treatment arms, our results for the observed decline in essentialism among treatment respondents with high genetic knowledge are fairly robust given that they are supported in the sensitivity analysis. While the observed increase in essentialist belief among treatment respondents with no genetic knowledge does not retain significance when 'no' and 'low' knowledge categories are grouped together, this suggests that changes may be specific to those with no genetic knowledge, an effect attenuated when combined with those with low knowledge. That a significant change from pre-test to post-test is observed among treatment respondents with 'no knowledge' in the analysis using the 4-category variable, despite a small number of cases, is noteworthy and warrants further attention.

This graph plots the interaction of genetic knowledge measured as a 4-category ordinal variable and the study arm allocation group variable. It shows the predicted average change in the pre-test and post-test genetic essentialism scores of each of the four genetic knowledge groups within the control and treatment groups separately. We used Stata's "margins" command to produce the graph.

Finally, in additional non-experimental analyses examining whether the specific results test-takers receive affect their belief in genetic essentialism, we focused on the treatment group only. Descriptive statistics (S7 Table) and findings from OLS regressions (S12–S14 Tables) are shown in the supplementary information. Most respondents (96%) listed a specific European ancestry, and 92.3% "confirmed" a known European ancestry with the test results. Fewer (15.9%) listed a specific non-European ancestry but only 1.1% "confirmed" it. A substantial majority received results reporting a new European ancestry (91%), while 61% "discovered" a new non-European ancestry (S7 Table).

However, we did not find any significant effects on treatment respondents' essentialist beliefs from these specific test results or their interaction with genetic knowledge. Although "confirming" a non-European ancestry reduces genetic essentialism in S13 Table, Model 1 (controlling for demographic variables) & Model 2 (controlling for demographic variables and genetic knowledge), this effect disappears once interactions between "confirmed" non-European ancestry and genetic knowledge are added in Model 3, reinforcing the role of genetic knowledge identified above. Further, the small number of treatment respondents "confirming" non-European ancestry does not offer confidence that this is a meaningful effect. Neither "confirmation" of a known ancestry (S12 and S13 Tables) nor "discovery" of a new one (S14 Table) from an admixture test significantly influenced genetic essentialism, supporting our hypothesis about null main effects. However, our hypothesis of an interaction between these test results and genetic knowledge was not supported.

## Discussion

This study analyzed whether taking GATs increases or decreases genetic essentialist beliefs about race, and whether changes in essentialism differ by baseline genetic knowledge. To our knowledge this is the first study to examine the effect of taking GATs on racial essentialism using an RCT, which is critical for separating the tests' effects from the motivations of the people who buy them. Test-takers may purchase these tests for reasons related to their racial beliefs and ideologies; for White nationalists, GATs may be a means to validate racial purity [52], while progressive White people may seek racial mixture [24], approaching these tests as "methods for performing racial harmony and assuaging white guilt" [53]. Without randomization, such motivations from test-takers would bias the findings of the tests' effects. Further, a

control group permits us to test internal validity; amidst growing media exposure about GATs and the genomic revolution [17,33], increases in genetic essentialism overall could reflect general changes in public views over time rather than the testing experience.

While our primary analysis testing whether taking GATs impacts individuals' concepts of race showed no difference in effect between treatment and control arms, secondary analyses suggest the potential that taking the tests did not uniformly increase or decrease test-takers' belief in racial essentialism. Instead, among the test takers in the treatment arm, changes in essentialist views were conditional on different levels of genetic knowledge. Those with high genetic knowledge taking GATs decreased their genetic essentialism. Results suggest that test-takers with very limited genetic knowledge may increase their belief in genetic essentialism, although additional research should explore this finding further.

While we did not randomize participants according to their baseline levels of genetic knowledge, these secondary analyses suggest that genetic knowledge is a potentially important modifier of the impact that genetic testing has on essentialist views. Having sufficient genetic knowledge to interpret the results may be a critical precursor to how genetic essentialist views change after taking a test. While many DTC testing companies provide technical scientific descriptions of the tests on their websites, they rarely provide a clear explanation about the relationship between genes and race or offer assistance on interpretation. As a result, many test-takers have trouble comprehending their results or translating the technical, scientific information into meaningful information for their genealogy and identity [1]. Indeed, treatment participants often expressed confusion over how to make sense of test results, both immediately after viewing them and in the post-test survey. In open-ended comments, one respondent wrote: "I need narrative results, not just a chart with numbers. I consider myself pretty intelligent but honestly the numbers and results made very little sense to me." Others noted it would take too much time to read up and fully understand them. Individuals who lack a foundation in basic genetics may be less likely to critically evaluate the tests' limitations. With admixture tests' biogeographical categories mapping onto commonly used racial groups, the tests appear to tell test-takers their race. This superficial interpretation may be the default for those who lack the required knowledge to engage with the results and not make erroneous inferences.

While our secondary analyses focused on the role of genetic knowledge on changes in essentialist beliefs between the pre-test and post-test, genetic knowledge is also inversely associated with people's baseline essentialist beliefs before testing. In the pre-test survey, those with higher genetic knowledge had lower genetic essentialism scores (shown on the left axis of panels in Fig 3). For those with high genetic knowledge, the testing experience provides an opportunity to reflect critically on what these tests purport to show. Taking a test thus has a polarizing effect, magnifying differences in essentialist beliefs even further between those with weaker and stronger understandings of the science behind them.

Despite the low number of observations of test-takers with very limited genetic knowledge, we believe the observed increase in genetic essentialism is important to report and should be followed up. As the first RCT in the area of genetic ancestry testing and genetic essentialism, results from this study are valuable both for the robust significant findings they provide and also in shaping an agenda for future research. We hope that the observed contrast between the 'no knowledge' and 'high knowledge' groups will instigate further research, and a sample with a greater number of observations with limited genetic knowledge can further test whether limited genetic knowledge plays a magnifying role in increasing genetic essentialism as a result of taking GATs. We also believe this study provides a rationale for future RCTs that randomize test-takers based on differing levels of genetic knowledge. Our secondary analysis provides

indication of potential moderation effect by genetic knowledge. A formal moderation analysis in future research might confirm this signal of an effect.

There are other limitations to our findings. First, the study described here focused only on native-born White Americans. Furthermore, because our target population was people willing to take GATs, it does not represent all native-born American Whites; however, large proportions of people state their willingness to take free GATs [54]. Second, the questions that tested genetic knowledge were limited in number. As our findings suggest that genetic knowledge may play an important role as a potential moderator of GATs' effect on essentialism, future researchers will benefit significantly from incorporating more developed scales of genetic knowledge, such as that developed by Jallinoja and Aro [55]. We expect that future studies incorporating such scales will show an even more robust effect than that found here. Third, we used one company for the tests, while DTC companies differ in the way their tests are analyzed, presented, and marketed, as well as their willingness to interpret individual findings or provide genetic counselling [56,57]. However, we note that among the largest DTC companies, presentation and interpretation services are comparable, and we expect that genetic literacy will be a major mediator in interpreting the results in all DTC companies that do not offer genetic counselling.

This is a study of a randomized sample–to eliminate the effect of consumers' motivations for buying the tests. Our findings do not necessarily apply to the self-selected people who currently purchase GATs, but speak to reactions of average individuals who are willing to take the tests but do not buy them. Yet sizable numbers of people receive this information without seeking it. Many receive tests as gifts or take them at a relative's request. Others are primarily interested in health-related tests and receive genetic ancestry information incidentally. Others are tested for television programs, company promotions, or research [45,58]. Personal genomic information may become more widely available on a societal level [25] and if so, our findings may be relevant for an even larger population.

Qualitative research shows that those who purchased the tests to aid their genealogical pursuits were highly motivated to make sense of their results and devoted considerable time to learning the science of population genetics. In contrast, those who purchased the tests for other reasons–to uncover or confirm ancestral identities or out of curiosity–typically were less motivated to learn about the science behind them and sometimes misinterpreted their results [24]. Thus, even those who purchase the tests may vary in their levels of genetic knowledge, understanding of the results, and motivations to educate themselves.

Another important avenue for future research is to determine whether actively increasing test-takers' genetic literacy will reduce their belief in genetic essentialism, even for those initially holding essentialist beliefs. We examined the impact of higher and lower starting points in genetic knowledge, but did not manipulate genetic knowledge itself, for example by teaching respondents more about genetics. People with higher pre-test genetic knowledge may also have other characteristics that contribute to the testing experience's reduction in essentialism such as openness to new ideas or stronger critical thinking. Yet, just as teaching adolescents scientifically accurate information about genetic variation between and within races can reduce their racial biases [59], increasing any test-takers' genetic knowledge may also help them put their test results in context. Such a finding might imply that educational materials or online genetics modules for future test-takers could help prevent these tests from advancing historically destructive views.

The popularity of GATs has generated considerable debate over how taking these tests will affect popular belief in racial essentialism. Some speculated that taking the tests would increase essentialist beliefs while others argued that the testing experience might decrease essentialism. We found that both processes appear to be relevant simultaneously, for different respondents.

Because sub-group effects in different directions can cancel each other out, examining average effects alone risks drawing the mistaken conclusion that these tests do not affect essentialism. This study also points to the importance of respondents' genetic knowledge in the testing process. Genetic ancestry testing appears to polarize test-takers by strengthening the beliefs about racial essentialism that they already held. Given the significant negative repercussions of essentialist beliefs about race [12–14,16–18], the reinforcement of those beliefs–even for some test-takers–should be a cause for concern.

## Supporting information

**S1 Appendix.**
(DOCX)

**S1 Fig. Example of admixture test results.** Figure Credit: Family tree DNA.
(DOCX)

**S2 Fig. Example of mtDNA test results, haplogroup migration map.** Figure Credit: Family tree DNA.
(DOCX)

**S3 Fig. Pre-test and post-test genetic essentialism by genetic knowledge (dichotomous) for control (N = 419) and treatment groups (N = 375).** This graph plots the interaction of genetic knowledge measured as a dichotomous variable and the study arm allocation group variables. It shows the predicted average change in the pre-test and post-test genetic essentialism scores of those with lower and higher genetic knowledge within the control and treatment groups separately. We used Stata's "margins" command to produce the graph.
(TIF)

**S1 Table. Odds ratios of remaining in study (N = 995[a]).**
(DOCX)

**S2 Table. Baseline characteristics of all participants, those lost to follow-up, and those remaining in the study.**
(DOCX)

**S3 Table. Comparison of study sample to American Community Survey (ACS) sample and survey on Genomics Knowledge, Attitudes and Policy views (GKAP) sample.**
(DOCX)

**S4 Table. Genetic Essentialism Scale for Race (GESR).**
(DOCX)

**S5 Table. Genetic knowledge questions, frequencies, and point values (N = 802).**
(DOCX)

**S6 Table. Genetic knowledge scale distribution in raw scores, 4-category ordinal measure, and dichotomous measure.**
(DOCX)

**S7 Table. Descriptive statistics for treatment respondents' test result variables (N = 377).**
(DOCX)

**S8 Table. Mixed model results showing pre- and post-test differences in genetic essentialism between the control and treatment groups, disaggregated by genetic knowledge**

**categories.**
(DOCX)

**S9 Table. Pairwise contrasts of genetic essentialism scores across the two waves for groups with different genetic knowledge levels (4-category ordinal measure) (N = 794).**
(DOCX)

**S10 Table. OLS regression on post-test genetic essentialism (N = 794).**
(DOCX)

**S11 Table. Pairwise contrasts of genetic essentialism scores across the two waves for groups with lower and higher genetic knowledge levels (dichotomous measure) (N = 794).**
(DOCX)

**S12 Table. OLS regression of European ancestry "confirmed" on treatment respondents' post-test genetic essentialism (N = 360).**
(DOCX)

**S13 Table. OLS regression of non-European ancestry "confirmed" on treatment respondents' post-test genetic essentialism (N = 60).**
(DOCX)

**S14 Table. OLS regression of "discovery" of ancestry on treatment respondents' post-test genetic essentialism (N = 375).**
(DOCX)

## Acknowledgments

The authors would like to thank William C. Carlquist, Marcella Chan, Mesmin Destin, Meredith Dost, Greg Duncan, Steven Heine, Torsten Heinemann Voigt, Jennifer Hochschild, Arne Kalleberg, Catherine Lee, Ann Morning, Alondra Nelson, Ian Tietjen, Jay Van Bavel, and the Russell Sage Foundation.

## Author Contributions

**Conceptualization:** Wendy D. Roth.

**Data curation:** Wendy D. Roth, Kaitlyn Jaffe.

**Formal analysis:** Wendy D. Roth, Şule Yaylacı, Kaitlyn Jaffe.

**Funding acquisition:** Wendy D. Roth.

**Investigation:** Wendy D. Roth, Kaitlyn Jaffe.

**Methodology:** Wendy D. Roth, Şule Yaylacı.

**Project administration:** Wendy D. Roth, Kaitlyn Jaffe.

**Resources:** Wendy D. Roth.

**Software:** Wendy D. Roth, Şule Yaylacı.

**Supervision:** Wendy D. Roth, Şule Yaylacı.

**Validation:** Wendy D. Roth, Şule Yaylacı.

**Visualization:** Şule Yaylacı.

**Writing – original draft:** Wendy D. Roth, Şule Yaylacı, Kaitlyn Jaffe.

**Writing – review & editing:** Wendy D. Roth, Şule Yaylacı, Kaitlyn Jaffe, Lindsey Richardson.

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
