## [Decision Letter · Decision Letter 0]

21 Oct 2019

PONE-D-19-21371

Do genetic ancestry tests increase racial essentialism? A randomized controlled trial shows it depends on genetic knowledge

PLOS ONE

Dear Dr Roth,

Thank you for submitting your manuscript to PLOS ONE. After careful consideration, we feel that it has merit but does not fully meet PLOS ONE’s publication criteria as it currently stands. Therefore, we invite you to submit a revised version of the manuscript that addresses the points raised during the review process.

The reviewers both had generally very positive comments about this article. However, they highlighted a number of issues that you should address. 

We would appreciate receiving your revised manuscript by Nov 15, 2019. To enhance the reproducibility of your results, we recommend that if applicable you deposit your laboratory protocols in protocols.io, where a protocol can be assigned its own identifier (DOI) such that it can be cited independently in the future. For instructions see: http://journals.plos.org/plosone/s/submission-guidelines#loc-laboratory-protocols

A rebuttal letter that responds to each point raised by the academic editor and reviewer(s). This letter should be uploaded as separate file and labeled 'Response to Reviewers'.A marked-up copy of your manuscript that highlights changes made to the original version. This file should be uploaded as separate file and labeled 'Revised Manuscript with Track Changes'.An unmarked version of your revised paper without tracked changes. This file should be uploaded as separate file and labeled 'Manuscript'

We look forward to receiving your revised manuscript.

Kind regards,

Mellissa H Withers, PhD, MHS

Academic Editor

PLOS ONE

**Journal Requirements:**

2. We note that Figure 1 in your submission may contain copyrighted images. All PLOS content is published under the Creative Commons Attribution License (CC BY 4.0), which means that the manuscript, images, and Supporting Information files will be freely available online, and any third party is permitted to access, download, copy, distribute, and use these materials in any way, even commercially, with proper attribution. For more information, see our copyright guidelines: http://journals.plos.org/plosone/s/licenses-and-copyright.

a) You may seek permission from the original copyright holder of Figure(s) [#] to publish the content specifically under the CC BY 4.0 license.

**Comments to the Author**

1. Is the manuscript technically sound, and do the data support the conclusions?

Reviewer #1: Yes

Reviewer #2: Yes

2. Has the statistical analysis been performed appropriately and rigorously? 

Reviewer #1: Yes

Reviewer #2: Yes

3. Have the authors made all data underlying the findings in their manuscript fully available?

Reviewer #1: Yes

Reviewer #2: Yes

4. Is the manuscript presented in an intelligible fashion and written in standard English?

Reviewer #1: Yes

Reviewer #2: Yes

5. Review Comments to the Author

Reviewer #1: I found this article very interesting and well written. A randomized trial to assess the effect of genetic ancestry testing on racial essentialism was much needed and the results are important.

My only concern is that the secondary analyses evaluating the modifying effect of genetic knowledge on essentialism after the test can only be considered suggestive and should be presented more cautiously. The reason for my concern is that as the authors clearly state in the discussion section, they did not manipulate genetic knowledge experimentally and therefore there could be other reasons why they see the modifying effect of genetic knowledge (i.e. openness to new ideas or stronger critical thinking).

My recommendation is that they present the main results of the randomize trial, and then clearly state that they conducted secondary analyses within the test group to explore possible modifiers that might be interesting to test in additional randomization experiments. The current title, for example, could be misleading, since it states: “A randomized trial shows that it depends on genetic knowledge”. I do not think that the secondary analysis with lack of randomization of genetic knowledge can be conclusive in this respect. I do think is interesting and should be further explored in future research projects. I would be very interested in reading about a trial that randomizes genetic knowledge among people with the same level of education and social background.

Reviewer #2: Summary:

Roth and colleagues conducted a randomized controlled trial to understand the effects of genetic ancestry tests on racial essentialism. Interestingly, they find that the effect on racial essentialism depends on the participant’s level of genetic knowledge. Individuals with high genetic knowledge showed decreased racial essentialism, while individuals with low genetic knowledge showed increased racial essentialism after genetic testing. The study design is novel and the findings will contribute to the body of literature on the intersections of genetic science and society. However, several methods and results lack sufficient detail to evaluate the validity of the study conclusions.

Major comments:

1. Introduction: In their discussion of the meaning of race, the authors imply that the scientific community has concluded that race is purely a social construct. This is an oversimplified position that does not adequately reflect the complexity of race. While race is primarily a social construct, to say there are no biological or genetic differences between different ancestral groups is incorrect. In biomedical contexts, race can still be an important variable to consider. A better way to counter genetic essentialism is not to completely deny the existence of genetic differences, but to emphasize that genetics is not the only factor that influences race, and that these genetic differences do not give certain races superior abilities. Suggested references to improve the authors’ discussion of race are Ifekwunigwe et al. (PMID: 30078844), Wagner et al. (PMID: 27874171), Burchard et al. (PMID: 12646676), and Risch et al. (PMID: 12184798).

2. Participants and procedure. Other types of direct-to-consumer genetic testing are available, though less common. It seems that providing the control group with some type of non-ancestry genetic results would have been a better comparison than giving them no genetic testing at all. Could the authors comment on this?

3. Genetic essentialism scores: Both the introduction and methods reference the novelty of the genetic essentialism scale developed by the authors. Has this scale been previously validated?

4. Measurement of genetic knowledge. The authors measure genetic knowledge using two questions from the Survey on Genomics Knowledge, Attitudes and Policy Views (GKAP). The rationale for using these two questions alone is unclear. Can the authors justify why they selected this survey rather than using validated surveys for genetic knowledge, such as the one developed by Jallinoja and Aro (PMID: 10848030)?

5. Calculation of genetic knowledge score. The authors calculate a genetic knowledge score weighting each question by the percent of study participants who responded correctly. However, knowledge levels of study participants may not be representative of the larger general population. Could the authors use weights from the original GKAP survey to confirm they are not over- or under-estimating the levels of genetic knowledge of their study participants? A comparison of the percent answering correctly in this study vs. the original GKAP would also be informative regarding the overall genetic knowledge level of this study population.

6. Analyses. The thorough attrition pattern analysis was valuable and a great addition to the paper.

7. Table 1. This information is redundant with what is presented in Figure 3. Table 1 could be moved to the Supplement for those interested in the exact numbers. Please also add sample sizes of each knowledge level in the control and treatment groups to this table.

8. Representativeness of study population. How representative is the study population of the U.S. non-Hispanic white population? The authors describe this as their goal during the study sampling, but never present information regarding how successful their sampling strategy was.

9. Supplemental Table 6. This table is the only presentation of study population characteristics, but it is confusing. All variables are presented with a mean and standard deviation, even for variables where these are meaningless metrics (eg, sex). For the categorical variables, please present the number of participants in each category and the frequency instead. Furthermore, please add a brief description of the population characteristics in the main text of the paper to provide context for the findings.

Minor comments:

1. Measures. Please provide a citation for the GKAP.

2. Statistical Analysis, Line 200. The authors use the abbreviation OLS, but do not define this abbreviation until line 221. Please provide abbreviation the first time it is used in the text.

3. Analyses. The authors mention that their models adjust for living in the South. How was geographic region defined?

4. Figures 2 & 3. These figures need more detailed figure legends. Figure 2 contains very little information of value and could be omitted or moved to the Supplement.

5. Supplementary Tables. Several of the supplemental tables present results for 3 models. However, the details of these models are not clearly described. Models 1 and 2 are also referenced in line 265 of the text without explanation of what these models contain. Clearer definitions of the models would be helpful. A footnote should be provided with each table listing the variables included in each model to make it easier for the reader.

6. Tables and Figures. Throughout the paper, the authors refer to figures and tables with the title followed by the number (eg, Fig 3), except when referring to Supplemental tables, when they list the table number first (eg, S5 Table). The Supplemental Tables should follow the same pattern as the figures and tables in the main text (eg, Table S5 or Supplementary Table 5).

6. PLOS authors have the option to publish the peer review history of their article (what does this mean?). If published, this will include your full peer review and any attached files.

Reviewer #1: No

Reviewer #2: Yes: Melinda C. Aldrich, Victoria L. Martucci

---

## [Author Response · Author response to Decision Letter 0]

19 Nov 2019

We would like to begin by sincerely thanking the reviewers for their detailed and helpful comments. The reviewers offered valid critiques, which have led us to rethink and revise many aspects of the paper. We feel that the paper is much improved as a result of the reviewers’ comments, and are very grateful for the time and effort that they put in to helping us improve it. 

We are also extremely grateful for the many positive and encouraging comments from all the reviewers. We truly appreciate that you have found value in the paper and that you took the time to express it.

Below we address the comments of the reviewers and the Editor. We have grouped the comments thematically. The line numbers refer to the version with track changes removed.

Genetic Knowledge as a Modifying Effect

Reviewers 1 and 3 raised concerns about the presentation of the analysis of genetic knowledge. Reviewer 1 noted that because we did not manipulate genetic knowledge experimentally, we should emphasize that these findings are suggestive and they should be presented more cautiously. Similarly, Reviewer 3 cautioned against describing our findings as a moderation. In line with the reviewers’ suggestions, we have reframed the paper to distinguish between our primary analysis, based on the randomization to treatment and control groups, and the secondary analyses, which are not. We have changed the title of the paper accordingly, and have reorganized the description of the analysis in the introduction as well as the hypotheses to sequentially describe primary and secondary analyses. We had previously used the term ‘moderation’, intending it as a synonym for the effect we observed in our interaction models in the Ordinary Least Squares (OLS) regression, and did not mean to imply that we had conducted a formal moderation analysis. To clarify this, we have reworded our language to avoid readers interpreting it as a moderation analysis. We also adjusted the wording of the secondary analyses to remove causal language, and edited the discussion section to avoid implying that the findings surrounding genetic knowledge are causally related to the testing intervention. 

Reviewer 3 also expressed concern about the structure of the genetic knowledge variable and the analytic approach used to test interactions between genetic knowledge and the study arm allocation group variable, particularly the comparison of the no, low, and medium knowledge groups to the ‘high knowledge’ group. The reviewer was concerned about the strength of the effect observed for the ‘no knowledge’ group as compared to ‘high knowledge’ group in the interaction model because the finding, the reviewer suggested, is based on about half the sample and because it is only significant at the .05 level. We appreciate the reviewer raising these concerns because it forced us to think carefully about this issue. First, we would like to point out that this comment seems to focus only on the OLS analysis (now S10 Table), which uses interactions between the genetic knowledge groups and assignment to the treatment group. However, we include the OLS analysis to lend further support to the mixed-effects models (S8 Table) and the post-estimation predictions that are based on them as presented in the pairwise contrasts (S9 Table). In particular, the pairwise contrasts show the predicted average probabilities of change in genetic essentialism within each genetic knowledge group between the pre-test and post-test surveys. Here, too, we see that the average predicted probability of treatment respondents in the ‘no knowledge’ group declined significantly between the pre-test and the post-test, and this is not simply relative to the ‘high knowledge’ group. 

 However, we acknowledge and agree with the reviewer that the finding is still based on a small number of cases in the ‘no knowledge’ group, and we have made several changes to be clearer about this and to test the robustness of our findings. As Reviewer 3 suggested, we did run the analyses with a dichotomized version of the genetic knowledge variable. We combined the ‘no/low’ knowledge categories and the ‘medium/high’ knowledge categories to create a binary measure of ‘lower’ and ‘higher’ genetic knowledge. With this binary measure, we find that the average predicted probabilities of genetic essentialism declined significantly in the treatment group with higher genetic knowledge (Δ = -0.025; p=0.007), while control respondents with higher genetic knowledge did not show a significant change (Δ = -0.010; p=0.280) (S11 Table and S3 Fig). This is in line with what we had found with the 4-category ordinal measure of genetic essentialism variable. We do not find a significant predicted change among treatment respondents with lower knowledge (Δ = 0.009; p=0.265), however. Thus, the dichotomous variable shows that our findings that high(er) genetic knowledge is predicted to decrease genetic essentialism after testing is robust; as expected, this analysis does not support the predicted increase in genetic essentialism among those with the least genetic knowledge because the ‘lower knowledge’ category is numerically dominated by those in the 4-category ‘low knowledge’ group.

 The reviewer also suggested using a continuous version of the genetic knowledge variable; however, we do not think a continuous version of this variable would be viable given that there are only four effective categories out of the six in the full range, as two of the categories have very few cases in them (S6 Table). When categories are few, it is not recommended to treat a categorical variable as continuous; the recommendation is to have at least 5 and preferably 7 categories to treat an ordinal variable as a continuous variable (Liddell and Krusche 2018). Furthermore, the variable is not normally distributed (see S6 Table). Also given that we are now aware of potential differences in our results based on different genetic knowledge levels, we do not think it is reasonable to assume that the one-unit effect will be the same across the different points of a continuous scale.

 We decided to include the analyses with both the 4-category ordinal and the dichotomous genetic knowledge variables in the paper. We believe that the preliminary indication of an expected increase in genetic essentialism for test-takers with no genetic knowledge is a noteworthy pattern that deserves further exploration. Our limitation is statistical power; in fact, it is noteworthy that we see a significant association despite the lack of statistical power. Because this is the first RCT of genetic ancestry testing on genetic essentialism (to our knowledge), we believe it is valuable to present the indication of a possible increase in genetic essentialism among those with the least genetic knowledge as something that might be interesting to explore in future research, while still using caution to avoid suggesting that this is a conclusive finding. We have edited our discussion to reflect this (lines 230-237; 257-258; 280-318; 369-373; 399-410). We have also added more information about these considerations in the supplement under the ‘Genetic Knowledge Variables’ section (p.5).

Representativeness of Sample

Reviewer 2 asked how representative the study population is of the U.S. non-Hispanic White population and said “The authors describe this as their goal during the study sampling, but never present information regarding how successful their sampling strategy was.” We apologize if our language was misleading, but want to clarify that the representativeness of the study population was not our goal. Our target population is not all native-born White non-Hispanics, but specifically native-born White non-Hispanics who are willing to take genetic ancestry tests (GATs) and have never received personalized genetic ancestry information. We expect this population to differ from the overall population of native-born White non-Hispanics, yet we were unable to find a representative sample of this particular target population before the study began. We have clarified this in the manuscript (lines 103-104; 150-152; 155-162). Nevertheless, we take the reviewer’s point that it is valuable for readers to know how this sample compares with the larger population of native-born White non-Hispanics. We have therefore added a table to the supplement (S3 Table) that compares our study sample to a representative sample of native-born White non-Hispanics from the 2015 American Community Survey. In the supplement (p.3-4), we discuss differences between our sample and this broader population, and note that these differences likely reflect patterns in who is willing to take genetic ancestry tests, as well as trends in who is willing to participate in research studies.

Reviewer 2 also asked how our sample’s knowledge of genetics compared to the larger general population. We therefore also added to S3 Table the responses to the survey items from the subsample of native-born White non-Hispanics in the Survey on Genomics Knowledge, Attitudes and Policy Views (GKAP). Again we compare our sample to this population, and note that those who are willing to take genetic ancestry tests (our target population) likely has higher genetic knowledge than all native-born White non-Hispanics.

Study Design

Reviewer 2 asked us to comment on why we chose to give our control group respondents no stimulus rather than some type of non-ancestry genetic test results. We made this decision because our goal is to simulate the effect of taking GATs relative to those who do not (as we now state on lines 106-107). We are interested in the potential effects of GATs on the population that takes them, rather than the relative effect of GATs vs. other genetic tests. To have compared those who take GATs to those who take health-based genetic tests, for instance, would be similar to more conventional non-inferiority clinical RCTs that test the equivalence of two treatments – whether one is inferior to the other for the treatment of a disease outcome. We expect that most people who do not take GATs likely do not take other genetic tests either. Therefore, we have maintained our control group based on the exploratory, innovative nature of the study and the fact that this type of RCT has not been conducted before (to our knowledge). Comparing the relative effect of different tests would be a reasonable next step after establishing the effect that concerns us here. 

Measures

Both Reviewers 2 and 3 asked for additional information about the essentialism scale. We added in more information about the development and validity of the scale. We also added in the Cronbach's alpha for this sample, and provided a clearer reference to the publication where the scale's development and validation are described in detail (lines 190-208).

Reviewer 2 noted that our models adjust for living in the South and asked how geographic region was defined. We use the same definition as the U.S. Census Bureau statistical regions. Specifically, the South census region includes Alabama, Arkansas, Delaware, the District of Columbia, Florida, Georgia, Kentucky, Louisiana, Maryland, Mississippi, North Carolina, Oklahoma, South Carolina, Tennessee, Texas, Virginia, and West Virginia. We added a description for how the geographic region 'South' was defined to the Supplement, under 'Control Variables' (pp.6-7).

Reviewer 2 also asked us why we developed the genetic knowledge scale out of the two questions from the GKAP survey rather than using validated surveys for genetic knowledge like the one developed by Jallinoja and Aro. Our answer is that when we began this study, we did not expect genetic knowledge to emerge as a major focus and predicted that educational attainment would capture the differences in respondents’ understandings of their GAT results. We view this as a limitation of the study, and believe that future studies need to build on this work by using a more robust scale. Although we had already mentioned this as a limitation of the study in our discussion section, we have further emphasized this point and have specifically mentioned the Jallinoja and Aro scale to direct potential future researchers towards it (lines 414-419).

Organization and Writing

In the Introduction, Reviewer 2 asked for a more nuanced discussion of the meaning of race that more adequately reflects its complexity. We thank the reviewer for pointing out the shortcomings of this section, as we did not intend to imply that the entire scientific community had concluded that race is purely a social construct or that it has no association with biological factors. We have edited this text (lines 49-56) to acknowledge this point and to clarify that socially constructionist views of race still acknowledge that those categories are partly based in biological or descent-based characteristics. We have also added in several additional references and thank the reviewer for providing them.

Reviewer 3 asked us to introduce earlier in the paper that we would examine specific ancestry results on GATs as an influence on genetic essentialism. We added this information to the abstract and also to the Introduction (lines 97-100).

Adding or Clarifying Statistics/Analyses

Based on Reviewer 2’s helpful comments, we have revised our presentation of study population characteristics. We added a brief description of the sample’s characteristics to the main manuscript (lines 166-171) to provide context for the findings. We also edited the previously-confusing supplement table (now S2 Table) that presented these descriptive statistics. It now shows frequencies for categorical variables and means and standard deviations only for continuous variables.

We have clarified our hypotheses both about the role of genetic knowledge, and about the “confirmation” and “discovery” of new ancestry. We had only included what we expect the low genetic knowledge to experience as a result of seeing the test results, which is an increase in genetic essentialism. To complement the section, we added the following sentence in lines 130-133: "By contrast, we expect test-takers with high knowledge of genetics to be more likely to read the results as evidence that races and racial traits are not determined solely by genetics, and thereby develop less essentialist views after testing." We also specified that with regard to the specific ancestry test results, we expect a null main effect, but that high genetic literacy will lead to less essentialism while low genetic literacy will lead to more genetic essentialism (lines 134-146).

At Reviewer 3’s suggestion, we clarified the claim, regarding the additional survey item measuring genetic knowledge, that “models that included this item showed similar results…” (now lines 220-221) to specify that these are models that we ran with this item. We decided not to include the statistics for those additional models, both because PLOS One does not allow footnotes and because including this information would require adding a somewhat lengthy explanation of how we coded genetic knowledge using three items rather than two. Because the third genetic knowledge item (how much of the genes of a Black person are the same as those of a White person) has considerable theoretical overlap with our dependent variable, we feel that adding a lengthy discussion of this does not add value to the paper. If the Editor feels it is more appropriate, we can simply delete this sentence.

We have edited our ‘Analysis’ section (lines 246-268) to be clearer about the statistical techniques we are using and which models they draw from. We specified “Ordinary Least Squares” the first time we provided the abbreviation, on line 249. We also clarified that the statistics procedure used for the second hypothesis (i.e., testing the effects of specific test results on genetic essentialism controlling for genetic literacy) is OLS regression. We say this on line 337 in the updated manuscript.

We confirmed that all our models include control variables. In the text, we stated this on lines 266-268 but changed the wording a bit to make it clearer: “All models, both experimental and non-experimental, control for living in the South, interaction with non-Whites and political party preference, as well as gender, age, and education.”

We thank Reviewer 3 for noting the omission of the statistic being reported in several places on the former page 10. The numbers refer to average predicted probabilities, also known as marginal effects. We changed the text by adding “predicted probabilities” in the necessary parts in the results section. We also would like to state that the average predicted probabilities capture the effect sizes that are relevant to our analysis.

We added information to the notes of S1 Table and S10 Table to explain what variables are added to each of these nested regression models. All variables in the model are shown in the left-hand column, so we did not add a footnote with this information. We also added information to the manuscript to explain what Models 1 and 2 of S13 Table are (now lines 344-345).

Figure Legends and Duplication of Information in Figures/Table

The reviewers noted that there was some duplication of information in the figures, text, and the table in the manuscript. However, they differed on what should be done about it. Based on our best judgment, we decided to move the table (formerly Table 1) to the supplement (now S9 Table). We also added the sample sizes of each knowledge level in the control and treatment group to this table. We have kept the changes in essentialism scores in the text because the exact numbers may not be clear from the figures. We added a note to Figure 3 and clarified what the values refer to in the text. We also made slight changes by bringing some details from the notes of Figure 2 to the main text (lines 273-266).

Minor Concerns

We did not intend to set up the expectation that we would be examining the impact of GATs on multiple types of essentialism. We edited the wording in the Introduction (lines 56-58) to clarify that here we were distinguishing between genetic essentialism and cultural essentialism or other types of essentialism that we do not study. We have also removed the names of the three sub-factors that contribute to the full genetic essentialism scale (lines 195-198). We wanted to indicate that this is a complex concept that a second-order model helps to capture, but removed these titles to avoid setting up readers to expect that we would be analyzing the sub-factors individually.

Reviewer 2 asked for a citation for the GKAP study. The Co-Principal Investigator sent us the reference that we include at the bottom of this memo. However, because PLOS One does not allow citations to unpublished works, we decided to instead cite a publication by the Co-PIs that uses the GKAP and discusses the survey details (line 213, citation 50). 

We have made all the minor editorial corrections pointed out by the reviewers, with the exception of referring to the Supplemental tables using the same pattern as for the main tables and figures (e.g. Table S1 instead of S1 Table). Here we are following the PLOS One formatting guidelines. 

Editors’ Comment

We have provided permission from the copyright holder to reproduce S1 Fig and S2 Fig.

Once again, we would like to thank the Reviewers for reading the paper so carefully and for their very helpful comments. We very much hope that the revisions to the manuscript adequately address the issues that were raised.

References:

Liddell, Torrin M. & John K.Kruschke. 2018. Analyzing ordinal data with metric models: What could possibly go wrong? Journal of Experimental Social Psychology 79: 328-348.

Survey on Genomics Knowledge, Attitudes, and Policies (GKAP), by Jennifer Hochschild and Maya Sen. GKAP 1 in 2011; GKAP 2 in 2017. GKAP 1 was funded by Robert Wood Johnson Foundation, as part of a Investigator Award in Health Policy Research, 2010-2013 (with Maya Sen). Surveys were conducted by Knowledge Networks (now GfK).

---

## [Decision Letter · Decision Letter 1]

19 Dec 2019

Do genetic ancestry tests increase racial essentialism? Findings from a randomized controlled trial

PONE-D-19-21371R1

Dear Dr. Roth,

We are pleased to inform you that your manuscript has been judged scientifically suitable for publication and will be formally accepted for publication once it complies with all outstanding technical requirements.

With kind regards,

Mellissa H Withers, PhD, MHS

Academic Editor

PLOS ONE

Additional Editor Comments (optional):

Reviewers' comments:

Reviewer's Responses to Questions

**Comments to the Author**

1. If the authors have adequately addressed your comments raised in a previous round of review and you feel that this manuscript is now acceptable for publication, you may indicate that here to bypass the “Comments to the Author” section, enter your conflict of interest statement in the “Confidential to Editor” section, and submit your "Accept" recommendation.

Reviewer #1: All comments have been addressed

Reviewer #2: All comments have been addressed

2. Is the manuscript technically sound, and do the data support the conclusions?

Reviewer #1: Yes

Reviewer #2: Yes

3. Has the statistical analysis been performed appropriately and rigorously? 

Reviewer #1: Yes

Reviewer #2: Yes

4. Have the authors made all data underlying the findings in their manuscript fully available?

Reviewer #1: Yes

Reviewer #2: (No Response)

5. Is the manuscript presented in an intelligible fashion and written in standard English?

Reviewer #1: Yes

Reviewer #2: Yes

6. Review Comments to the Author

Reviewer #1: Authors have addressed my concerns thoughtfully, as well as the concerns of other reviewers. I have no additional comments for the Authors.

Reviewer #2: Roth and colleagues present a significantly revised version of their manuscript regarding the effect of genetic ancestry testing on racial essentialism. The authors have made substantial changes to the manuscript, and we appreciate how thoroughly and thoughtfully they have addressed the comments raised. The revised manuscript is significantly improved. Overall, we feel the authors have addressed our concerns and recommend the manuscript be accepted for publication.

Comments:

1. Introduction: The authors have revised their discussion of the meaning of race to present a more nuanced view, as requested. These changes better represent the complexity of race and provide better context for the article.

2. Participants and procedure. We had questioned why the reviewers chose to provide no genetic tests to the control group. The authors have clarified this in their response and have incorporated language into the manuscript to emphasize that their goal was to compare people who receive genetic ancestry testing with those who do not. Their rationale that people who do not receive ancestry testing also rarely receive other genetic testing seems valid, and we agree that this work is an important first step in understanding the role of genetic ancestry testing on racial essentialism.

3. Genetic essentialism scores. In the original manuscript, it was unclear how the authors had developed the genetic essentialism scale and whether it had been validated. The authors have added more detail into the methods section and provided a reference to their previous paper where they developed the score. We appreciate the clarification and feel it appropriately addresses our concerns.

4. Measurement of genetic knowledge. We were unclear why the authors had used just a few questions from the Survey on Genomics Knowledge, Attitudes and Policy Views (GKAP) to measure genetic knowledge. However, the authors’ substantial revisions to the manuscript to emphasize that the genetic knowledge analyses were a secondary goal of the work and the revised title de-emphasizing the genetic knowledge results make this less of a concern. While better metrics of genetic knowledge would be valuable to this research, we understand that this was not the authors’ initial goal. The changes they have made to the manuscript and the addition of a more thorough discussion of measurements of genetic knowledge sufficiently address our previous concerns. We also appreciate the authors providing references to the GKAP to the Reviewers and to readers of the manuscript.

5. Figures and tables. We appreciate how carefully the authors have thought about which tables and figures to include in the revised manuscript. The revisions made to the study population characteristics table make it much easier to understand. We agree with the authors’ decision to move the original Table 1 to the supplement, as it was not adding much information beyond the main figures. We also appreciate the addition of sample sizes to this table. The updated figure legends also add much needed detail that was previously lacking.

6. Representativeness of study population. We thank the authors for adding comparisons between the study population and the overall non-Hispanic white population in both the demographics and genetic knowledge tables. The revised language around the study sampling goals and the added discussion regarding the representativeness of the study population address our previous concerns.

---

## [Editor Report · Acceptance letter]

3 Jan 2020

PONE-D-19-21371R1 

Do genetic ancestry tests increase racial essentialism? Findings from a randomized controlled trial 

Dear Dr. Roth:

I am pleased to inform you that your manuscript has been deemed suitable for publication in PLOS ONE. Congratulations! Your manuscript is now with our production department. 

With kind regards,

on behalf of

Dr. Mellissa H Withers 

Academic Editor

PLOS ONE